# Anti-Oxidative Therapy in Islet Cell Transplantation

**DOI:** 10.3390/antiox11061038

**Published:** 2022-05-24

**Authors:** Natsuki Eguchi, Kimia Damyar, Michael Alexander, Donald Dafoe, Jonathan R. T. Lakey, Hirohito Ichii

**Affiliations:** 1Department of Surgery, University of California, Irvine, CA 92697, USA; neguchi@hs.uci.edu (N.E.); kdamyar@uci.edu (K.D.); michaela@hs.uci.edu (M.A.); ddafoe@hs.uci.edu (D.D.); jlakey@hs.uci.edu (J.R.T.L.); 2Department of Biomedical Engineering, University of California, Irvine, CA 92686, USA

**Keywords:** islet cell transplantation, diabetes, oxidative stress, antioxidants, pancreatic β cell replacement therapy

## Abstract

Islet cell transplantation has become a favorable therapeutic approach in the treatment of Type 1 Diabetes due to the lower surgical risks and potential complications compared to conventional pancreas transplantation. Despite significant improvements in islet cell transplantation outcomes, several limitations hamper long-term graft survival due to tremendous damage and loss of islet cells during the islet cell transplantation process. Oxidative stress has been identified as an omnipresent stressor that negatively affects both the viability and function of isolated islets. Furthermore, it has been established that at baseline, pancreatic β cells exhibit reduced antioxidative capacity, rendering them even more susceptible to oxidative stress during metabolic stress. Thus, identifying antioxidants capable of conferring protection against oxidative stressors present throughout the islet transplantation process is a valuable approach to improving the overall outcomes of islet cell transplantation. In this review we discuss the potential application of antioxidative therapy during each step of islet cell transplantation.

## 1. Introduction

Type 1 diabetes (T1D) is a chronic autoimmune disorder characterized by the selective destruction of insulin-secreting pancreatic β cells. Diabetes has been linked extensively with secondary complications, including neuropathy, nephropathy, retinopathy, cardiovascular disease, peripheral vascular disease, and cerebrovascular accident. Currently, the standard treatment for patients with T1D is the administration of exogenous insulin. The tight glycemic control has shown to substantially reduce the severity of secondary complications; however, its tie to a severe incident of hypoglycemia makes it a lifesaving yet unsafe treatment [1]. 

In the select few T1D patients who show intractable impaired awareness of hypoglycemia, β cell replacement therapy is offered as an alternative therapeutic approach. Currently, in islet cell transplantation (ICT), although graft survival is reported to be at 80% five years post-transplantation (PT), insulin independence is found in only 44% of patients three years PT [2,3,4]. However, the outcome of ICT differs from center to center, and the Minneapolis team has reported the highest success rate, with 50% insulin independence in 25 patients five years PT [3]. Despite much improvement in recent years, currently, two–three pancreases may be required to achieve sufficient islet mass to achieve insulin independence in ICT because the majority of islets are lost during the isolation process and immediately after islet infusion. Antioxidative therapy to improve ICT has gained more interest in recent years as oxidative stress has been identified as an omnipresent stressor throughout the entire ICT process: isolation, purification, culture, infusion, and immunosuppressive therapy [5] (Figure 1). Therefore, the identification of antioxidants capable of conferring protection against oxidative stress is an important approach to improving overall outcomes of ICT [6]. In this review, we discuss the potential use of antioxidative therapy during each step of the ICT process to improve islet yield, viability, and function.

## 2. Islet Cells and Oxidative Stress

There has been a long held perception that pancreatic β cells are susceptible to oxidative stress due to their inherently low expression of antioxidant enzymes superoxide dismutase (SOD1/2), glutathione peroxidase(GPx), and catalase (CAT) [7]. However, more recently Stancill J. et al. demonstrated that when human EndoC-βH1 cells are exposed to physiologically relevant H_2_O_2_ levels (50 μM) in a continuous manner, β cells are able to detoxify it through the peroxiredoxin and thioredoxin antioxidant system [8,9,10]. Furthermore, in comparison to the low expression of GPx and CAT, peroxiredoxin, thioredoxin, and thioredoxin reductase genes are readily expressed in mice and rat β cells [8]. Thus, further studies in this area to identify highly expressed antioxidative enzymes in β cells will be crucial for the optimization of antioxidative therapy. 

The primary sources of ROS production in islet cells are thought to be xanthine oxidase(XO), cytochrome P450-based enzymes, reduced nicotinamide adenine dinucleotide phosphate (NADPH) oxidase, dysfunctional nitric oxide(NO) synthase, and mitochondrial dysfunction [5]. While ROS has generally been associated with imposing harmful effects on islet cells through negatively impacting cellular proliferation, survival, and the inducing signaling cascades to mediate cellular damage, it is not inherently a detrimental process [11]. For example, ROS has been shown to be essential for insulin signaling under basal conditions [12]. In mouse INS-1 cells, glucose treatment increased intracellular H_2_O_2_ and treatment with H_2_O_2_ scavengers inhibited glucose-stimulated insulin secretion [13]. However, induction of oxidative stress in mouse INS-1 cells also resulted in decreased glucose-stimulated insulin secretion (GSIS), suggesting that while ROS are necessary for GSIS, excessive ROS and oxidative stress impairs insulin signaling. Therefore, pancreatic β cells require a state of proper redox balance for optimal function, which is an important factor to consider for antioxidative therapy. In fact, several studies have suggested that reductive stress, which results from the accumulation of NADH such as in the case of excessive glucose metabolism, precedes oxidative stress in diabetes [14]. Considering that reduced glutathione, one of the primary ROS scavengers stimulated under antioxidative therapy, has been shown to induce reductive stress, further studies examining the connection between antioxidative therapy and reductive stress would be vital to improve overall islet cell transplantation outcomes [14]. For now, we focus on the oxidative stress and the application of antioxidative therapy during each step of ICT.

## 3. Antioxidative Treatment of Islets during Procurement and Preservation

The low content of antioxidant enzymes within the human pancreas, including SOD, makes the pancreas extremely vulnerable to reactive oxygen stress [6,15]. During pancreas procurement and cold ischemia during pancreas preservation, either prior to transplant or islet isolation, oxygen radicals are generated through the xanthine oxidase (XOD) [16] and NADPH oxidase pathways [17]. These pathways are activated during reperfusion and generate oxygen radicals too fast for endogenous SOD to cope [16]. 

The University of Wisconsin (UW) solution was the standard organ preservation solution for over 30 years in clinical preservation of the pancreas for transplantation. The UW solution has glutathione and allopurinol as reactive oxygen species (ROS) scavengers. Glutathione scavenges free radicals, ROS, and reactive nitrogen species (RNS) [18], while allopurinol inhibits XOD [19]. It was first used for kidney transplantation in 1988 [15], canine pancreas transplantation in 1989 [20], and human pancreas in 1990 with up to 24 h of cold preservation [21]. The UW solution was then used for the pancreas allocated for ICT with the release of the Edmonton protocol in 2006 [22]. 

Following the development of the UW solution, recent cold preservation formulations have been designed to address some of the limitations [23]. The UW solution is chemically unstable, with strict cold storage requirements, short shelf life, and high viscosity, making it difficult to flush organs during procurement [24]. The UW solution also inhibits enzymatic activity during islet isolation, including inhibitory effects from glutathione and allopurinol used as its antioxidants [25]. 

There has been considerable research and development into novel preservation solutions over the past decades, including Histidine-tryptophan-ketoglutarate (HTK) solution, also known as Custodiol, which is an intracellular crystalloid cardioplegic solution used for myocardial protection. Custodiol (HTK) was first used in the 2000s as an alternative for UW solution clinically [26], followed by the Institut Georges Lopez-1 (IGL-1) solution [27]. Between 1996 and 2005, the UW solution accounted for almost 60% of organs preserved, and HTK 11% [26]. IGL-1 is a newer solution that was first used for pig pancreas in 2014 [27], then for human pancreas in 2016 [28], although it was used for liver clinically in 2010 [29]. In comparison to the UW solution, HTK uses tryptophan, while IGL-1 uses glutathione and allopurinol (same as UW) as its ROS scavenger [30]. 

In a single-center, large-scale retrospective analysis of pancreas transplants (n = 252) using either UW or HTK, there was no demonstrated difference in outcome between the two groups [31]. This is supported by Indiana University pancreas transplant analysis, showing no clinically significant difference [32,33]. When tested for use in ICT, HTK and UW have the same preservation efficacy, but only on the pancreas with cold ischemia <10 h [34]. However, national registry data for pancreas transplants showed an increased incidence of detrimental effects and earlier graft loss after preservation with HTK compared to UW [35]. Custodiol (HTK) and its variation, Custodiol-N (composition provided in Table 1 [36]), are now being tested in a multi-prospective, randomized, single-blind, multicenter, phase III study on pancreas, liver, and kidney preservation prior to transplantation [36,37].

A meta-analysis of patients receiving a simultaneous kidney and pancreas transplantation, preserved using UW or IGL-1, showed that there is still a lack of data on the immediate results and no clinical studies on the long-term outcomes [30]. IGL-1 demonstrated good safety and efficacy in two case series for up to 17 h of preservation [28,38]. For the goal of supporting ICT, IGL-1 protected β cells yield better than both HTK and UW in preserving the pancreas from donors after controlled circulatory death (DCD III) [39]. 

More novel antioxidants have been tested on pancreas preservation and procurement, such as glutamine, melatonin, ascorbic acid, dimethyl fumarate (Tecfidera), and bardoxolone methyl analog (dh404). Glutamine, a natural antioxidant, has been shown to improve islet yield, viability, and endogenous pancreas glutathione level, if perfused into the pancreas prior to pancreatectomy both in rats [40] and in pigs [41]. Melatonin (N-acetyl-5-methoxytryptamine) has a strong antioxidant and anti-inflammatory effect. Melatonin is synthesized from tryptophan by the pineal gland, in addition to by ovaries, testes, bone marrow, gut, placenta, and liver [42]. In one research study, pig donors received melatonin and ascorbic acid (10 mg/kg intravenous) prior to pancreatectomy, while these antioxidants were added at 5 mM into the UW solution during pancreas cold storage. Compared to non-treated donors, melatonin reduced the oxidative stress markers malondialdehyde and 4-hydroxyalkenals during procurement, cold ischemia, and reperfusion. Meanwhile, ascorbic acid only partially reduced oxidative damage 30 min after reperfusion [43]. Dimethyl fumarate has also been shown to improve islet yield from rats if administered orally for two days prior to procurement and islet isolation [44]. Similar to DMF, dh404 treatment of Sprague–Dawley rats enhanced nuclear translocation of nuclear factor erythroid factor-related factor 2(Nrf2) and elevated heme oxygenase 1 (HO-1) expression in the pancreas and improved islet yield and viability [45]. Lastly, pretreatment of Sprague–Dawley rats with bilirubin also enhanced islet survival rate and viability through reducing lipid peroxidation [46]. At this time, none of these antioxidants are used clinically for pancreas procurement.

## 4. Antioxidative Treatment of Islets during Isolation and Processing

Following pancreas procurement and preservation, mechanically enhanced enzymatic digestion to dissociate the islets from their extracellular matrix makes them further susceptible to oxidative stress and cell death [47,48]. Few studies have investigated the impact of treatment with antioxidants during the isolation of islets on transplantation outcomes. Thomas et al. reported that supplementation of all the reagents used in the murine islet isolation process with 1 nM d-Arg-2’,6’-dimethyltyrosine-Lys-Phe-NH_2_ (SS-31)—a small water-soluble peptide—preserved mitochondrial polarization, diminished cell apoptosis, and enhanced islet yield [47]. Additionally, Bottino et al. demonstrated that the activation of nuclear factor-κB (NF-κB) and poly (ADP-ribose) polymerase (PARP)—the two critical signaling pathways responsible for inducing oxidative impairment—emerges during the isolation of insulin-secreting cells. They further showed that the addition of 34 μM manganese [III] 5,10,15,20-tetrakis [1,3-diethyl-2imidazoyl] porphyrin (MnTDE)—a catalytic antioxidant—to the isolation medium decreased NF-κB signaling and PARP activation in human islets [48]. Finally, the addition of glutathione ethyl-ester (GEE) during the distention and digestion of murine islets has been reported to reduce the intracellular ROS contents of the islets and attenuate islet cell apoptosis, allowing the maintenance of their viability [49]. Despite these promising benefits, more evidence is required to show efficacy for the widespread clinical applications of antioxidants used in islet cell isolation to improve transplantation outcomes. 

## 5. Antioxidative Treatment of Islets during Cell Culture

To disrupt post-isolation oxidative stress and minimize cell loss before transplant, supplementing the islet cell culture media with exogenous antioxidants such as manganese superoxide dismutase (MnSOD) is an attractive area of study [50]. Incubation of human islet cell culture media with 1 nM SS-31 peptide for 72 h decreased islet cell apoptosis and moderately increased islet viability. SS-31 is a small permeable peptide with the capacity to accumulate in the inner mitochondrial membrane, act as a scavenger of ROS at their site of origin, and ultimately prevent cell apoptosis [47,51]. Moreover, treating human islet culture media with 34 μM MnTDE for 60 h improved insulin secretion and islet survival [48]. Thus, MnTDE has been found to exert protective effects on islet cells when added to the isolation and culture media.

Curcumin, a hydrophobic polyphenol derived from the rhizome of the herb *Curcuma longa*, has also been studied during the past few years as an antioxidant as well as an anti-inflammatory and anti-cancer reagent for the treatment of different types of cancer, diabetes, autoimmune disorders, and cardiovascular diseases [52,53,54,55,56,57,58,59,60]. A study revealed that the supplementation of murine islet culture media with curcumin significantly decreased the secretion of monocyte chemoattractant proteins (MCP-1) compared to untreated islets [61]. The NF-κB pathway regulates the expression of MCP-1 in human β cells, and it has been suggested that prolonged periods of insulin independence in post-transplant T1D patients are associated with low levels of MCP-1 in β cells [62,63]. Despite the therapeutic effects of curcumin, the hydrophobic nature of the molecule requires the utilization of toxic organic solvents, such as alcohol or dimethylsulfoxide (DMSO), for solubilization. Therefore, it is necessary to investigate alternative solubilization approaches to eliminate the need for toxic organic solvents and develop curcumin’s clinical applications. For instance, peptide micelle-mediated delivery of curcumin with R3V6 established the peptide’s ability as an appropriate carrier that also preserves islet viability after transplantation [64]. 

Furthermore, a recent study has shown the antioxidative and protective effects of tetrahydrocurcumin (THC), one of the major metabolites of curcumin with a more substantial antioxidative power. Supplementing murine islet culture media for 24 h with THC resulted in a 1.3-fold increase in glucose-induced insulin release compared to islets without THC. Additionally, THC treatment of the culture media attenuated cytokine-induced damage associated with cell apoptosis. These findings indicate the strong antioxidative capacity of THC to enhance islet cell function and graft survival PT [65]. 

Glutathione, a tripeptide synthesized from glutamate, glycine, and cysteine, also indicated encouraging results as an antioxidant in modulating the pro-inflammatory state of human islets. The treatment of islet cell culture media and incubation for 48 h with glutathione significantly reduced the production of MCP-1 in human islets compared to the untreated group, indicating this antioxidant’s ability to protect the islets and suppress the inflammation during engraftment [66,67]. Another group evaluated the antioxidative strength of GEE, the esterified form of glutathione. When human islets were cultured for 24 h in 20 mM GEE supplemented media, they had a significantly lower apoptosis rate than islets incubated in standard CMRL media. The esterification of glutathione by an ethyl-ester to form GEE increases its bioavailability and converts it into a lipid-soluble molecule that can cross the cell membrane and enter the cells [49,68]. Therefore, the observed improvement in human islet cell viability and the enhanced bioavailability of GEE makes it an attractive antioxidant scavenger for further clinical islet transplantation studies. 

In summary, the activation of NF-κB and PARP pathways are major contributors to potential oxidative damage during the isolation of insulin-secreting cells. The antioxidants and SOD mimetics employed during islet isolation and culture can target these pathways, suppress the activation of NF-κB, diminish the secretion of MCP-1 and interleukin (IL)-6 by human β cells under stress, and reduce the release of ROS by macrophages. Managing oxidative stress during isolation and culture can substantially impact islet yield, viability, and long-term graft function PT [48,50,69,70]. Many reagents have demonstrated strong antioxidative capacity in vitro and in vivo; however, their clinical applications have been limited due to a lack of data to establish their safety and efficacy in human islet transplantation procedures. Among the discussed antioxidants, MnTDE, GEE, and SS-31 appear to be ideal candidates for further investigations as they have shown positive impact when utilized during both the isolation and culture of pancreatic islets. 

## 6. Antioxidative Treatments during Islet Cell Infusion

During the infusion and immediate post transplantation phase of the islet, islet cells must overcome multifaceted challenges with hypoxia, hypoxia/reperfusion injury, and immediate blood-mediated immune response (IBMIR). Islet cells only account for 1% of the pancreas mass but receive approximately 10% of the total arterial blood delivered to the pancreas [71]. Furthermore, an islet cluster consists of around 1000–2000 cells per islet cluster. Keeping in mind that the revascularization of transplanted islets starts 2–4 days after transplant and requires 10–14 days to complete, it is no surprise that central necrosis is commonly detected in transplanted islets [72]. Hypoxia/reperfusion induces necrosis and cell apoptosis by releasing Caspase 3 and stress-related apoptotic factors, production of ROS due to inflammation, and activation of JAK/STAT, JNK/p38, and NF-κB pathways [73,74,75,76]. In addition to hypoxia, IBMIR in pancreatic islet cell transplantation begins immediately after islet infusion and peaks within three hours [77]. It is characterized by platelet consumption, activation of the coagulation and complement cascade, and leukocytic infiltration, resulting in significant inflammation and destruction of islet tissue immediately after transplantation [78,79]. IBMIR represents a major hurdle for transplant, and it is currently being targeted by NF-κB and JNK phosphorylation inhibitors, anticoagulants, complement inhibitors, and antioxidant administration to reduce inflammation [80,81,82]. Hypoxia and IBMIR are tightly intertwined. Inflammatory cytokine and ROS production as a consequence of the islet isolation process and hypoxia results in the release of damage-associated molecular patterns (DAMP), which acts as a chemoattract for immune cells and thus activate the innate immune response and subsequently IBMIR [83]. Therefore, targeting hypoxia through attenuation of oxidative stress and improving vascularization are vital to enhancing the viability of islet cells during the immediate PT period. 

Several antioxidative agents have been shown to confer protection against hypoxic stress. In vitro, resveratrol and nobiletin, both of which are safe for humans, have been shown to reduce ROS production and attenuate apoptosis under hypoxic conditions. Furthermore, resveratrol and nobiletin increased insulin and C-peptide secretion, suggesting improved survival and function of the islet cells [84,85]. Importantly, this was associated with increased secretion of vascular endothelial growth factors (VEGF), a signal protein that stimulates the formation of blood vessels [84]. In vivo, resveratrol treatment increased the vascularization of islet cells transplanted into the kidney capsule of streptozotocin (STZ)-induced diabetic mice and concurrently improved glucose tolerance and increased insulin stained area [86]. Bilirubin, an antioxidant product of HO-1, decreased oxidative stress and death in murine islets under hypoxic stress in vitro [83,87]. In vivo, bilirubin administration of STZ-induced diabetic rats with intraportal islet cell transplantation improved glucose tolerance and reduced serum inflammatory cytokine. Notably, bilirubin-treated diabetic rats transplanted with a suboptimal dose of islet cells (700 IEQ or 500 IEQ) exhibited significantly lower fasting blood glucose levels compared to the vehicle-treated group, further supporting the protective effects of bilirubin in islet cell transplantation [88,89]. These results were associated with significant reduction in inflammatory cytokines, including interleukin 1 β (IL-1β), tumor necrosis factor-α (TNF-α), MCP-1, and NO, suggesting additional anti-inflammatory properties of bilirubin. Furthermore, bilirubin also increased vascularization, as evidenced by increased blood vessel formation in ε-polylysine-bilirubin conjugate (PLL-BR) encapsulated islet cells transplanted into diabetic mice compared to control [90]. Unfortunately, its clinical applications have been limited due to its insolubility and short half-life. Several approaches have been taken to enhance drug delivery of bilirubin, such as encapsulation in pluronic F127-chitosan and other supramolecular carriers, and conjugation with hydrophilic polyethylene glycol (PEG) [83,91,92]. Several other antioxidants have been evaluated for improving islet survival during the immediate PT period. For example, epigallocatechin gallate (ECG), an Nrf2 activator, has been shown to protect rat islet cells from hypoxia reperfusion injury in vitro [93]. Pretreating islet cells with antiaging glycoprotein resulted in improved islet cell engraftment when transplanted into mice, an effect mediated by suppression of proinflammatory cytokines and chemokines [94]. Lastly, in vitro, exendin-4 protected mice islets against hypoxia-induced apoptosis and in vivo, exendin-4 treatment significantly improved survival of diabetic mice that received islet cell transplantation [95]. 

Lastly, various studies have recently explored the use of stem cell and stem cell-derived microvesicles to improve revascularization and viability in ICT. Treatment of porcine islets with human mesenchymal stem cell-derived exosomes under hypoxic conditions significantly decreased apoptosis and ROS production [96]. Furthermore, human islet cells treated with endothelial progenitor cell microvesicles (EPC-MV) exhibited increased development of capillary-like structures in vitro and in vivo. Interestingly, while rapamycin abrogated the angiogenic effect of growth factor-enriched medium on islet endothelial cells, the effect was not completely abolished when treated in EPC-MV, suggesting that EPC-MV induces angiogenesis through various pathways [97]. Stem cell-derived exosomes mediate angiogenesis by transferring pro-angiomiRNA and activating phosphatidylinositol-3-kinase (PI3K)/Akt, endothelial nitric oxide synthase (eNOS), and NF-κB signaling pathways in endothelial cells [97,98,99]. While the use of stem cell-derived microvesicles has gained interest in more recent years due to its several advantages over stem cell therapy, such as higher safety profile, lower immunogenicity, and the ability to cross biological barriers, several limitations, including the difficulty with characterization, pharmacokinetics and drug targeting, and an unknown safety profile hampers its application in clinical settings. 

## 7. Antioxidative Treatments in Islet Graft Recipients

After successful engraftment, transplanted islets face rejection and immunosuppressive medication toxicity challenges. The current immunosuppressive medication regimen follows a steroid-free protocol, commonly consisting of tacrolimus and rapamycin. However, immunosuppressors, such as tacrolimus and cyclosporin A, have been shown to cause β cell apoptosis and decrease insulin secretion [100]. Both tacrolimus and rapamycin have been shown to inhibit the mammalian target of rapamycin (mTOR) pathway, which has been suggested to be invaluable in maintaining β cell homeostasis and insulin secretion [101,102]. Importantly, the treatment of β cells (CRI-D2 β cell line) with tacrolimus decreased insulin secretion and viability, which was associated with a dose and time-dependent increase in ROS production, with a simultaneous decrease in antioxidative status [103]. Thus, several antioxidants, such as ginseng, gamma-aminobutyric acid (GABA), coQ10, antiaging glycopeptides, dipeptidyl peptidase-4 (DDP4) inhibitors, and exendin-4, have been evaluated for immunosuppressive medication toxicity and have shown to protect islets against tacrolimus-induced apoptosis through the attenuation of oxidative stress [94,104,105,106,107,108,109]. Therefore, the coadministration of antioxidants to combat the adverse effects of current immunosuppressive medication is a promising approach to target these issues. Favorably, several antioxidants have been identified as also exhibiting strong immunomodulatory effects: bilirubin, stem cell-derived EV, and DMF. Usage of these agents, in particular, would provide a double-edged sword against oxidative stress during the early PT period while providing effective immunosuppression for long-term management.

Stem cell extracellular vesicles (EV’s) and bilirubin have been most extensively studied for their use as an immunosuppressant for islet cell transplantation. In addition to suppressing the proliferation of PBMCs, stem cell EV and bilirubin have also been shown to be immunosuppressive through the induction of regulatory T cells. Exosomes of human bone marrow mesenchymal stem cell (MSC) cocultured with peripheral blood mononuclear cells (PBMC) successfully suppressed PBMC proliferation and induced regulatory T cells in the spleen of T1 diabetic mice transplanted with islet cells, rendering them entirely insulin-free without immunosuppressive medication [110]. The immunosuppressive effects of stem cell-derived EV may be mediated by its suppressive effects on macrophages, dendritic cells, and T cells. Recent studies have shown that stem cell-derived exosomes are capable of inducing phenotype conversion of macrophages into its anti-inflammatory M2 subtype through the horizontal transfer of miR146α and STAT3 [111,112,113]. M2 macrophages are known to secrete anti-inflammatory molecules, including IL-10, transforming growth factor β (TGFβ), and epidermal growth factor (EGF), some of which are responsible for inducing regulatory T cells. Furthermore, miR146α holds an essential role in preventing hyperactivity of CD4^+^ and CD8^+^ T cells and ensuring suppressive activity of regulatory T cells [114,115]. In terms of dendritic cells, MSC and MSC-EV/exosomes have been found to induce immunologic tolerance through the direct inhibition of effector dendritic cell function. In several in vitro studies, coculturing MSC-EV with monocyte-derived dendritic cells (moDC) attenuated their maturation through decreasing antigen uptake in immature dendritic cells. These moDCs also displayed low-level MHC Class II expression and increased secretion of anti-inflammatory cytokines, including IL-10 and TGFβ [116,117]. Moreover, MSC-EV cocultured with moDC expressed lower levels of CCR7, the receptor required for homing DC to secondary lymphoid organs, thereby limiting their ability to activate naive T cells. When MSC-EV-conditioned moDC were cocultured with naive T cells, they reduced T cell proliferation, induced higher expression of regulatory phenotype in T cells, and increased its secretion of anti-inflammatory cytokines (IL-10, IL-6, and TGFβ) [118]. Considering that in addition to its immunosuppressive properties, stem cell EV exhibit potent anti-inflammatory, antioxidative, and angiogenic effects, stem cell-derived EV therapy may represent a valuable approach to improving islet cell engraftment and viability during the early PT period and induce regulatory T cells for graft tolerance. 

Similar to stem cell-derived EV, besides demonstrating antioxidative and angiogenic properties, bilirubin exerts immunosuppressive properties by inducing regulatory T cells. In vitro, bilirubin treatment of CD-3 mAb-activated murine CD4^+^ T cells suppressed its proliferation by 52%. Additionally, cotreatment with bilirubin significantly suppressed activation-induced up-regulation of MHC Class II in both macrophages and dendritic cells, suggesting that bilirubin exerts multilevel immunosuppression through inhibiting both innate and adaptive immune cells [119]. Corroborating these findings, kidney capsule transplantation of ε-polylysine-bilirubin conjugate encapsulated islets in diabetic mice promoted the accumulation of M2 polarized macrophages in the islet graft [90]. Moreover, the treatment of C57BL/6 macrophage with bilirubin significantly increased surface expression of PDL-1, which has been shown to inhibit lymphocyte activation when bound to PD-1 and induce tolerance in islet allograft models [87,120]. Indeed, in vitro, macrophages exposed to bilirubin induced the formation of Foxp3+ T cells over a four-day culture with naive immune cells [87]. Lastly, mice transplanted with ε-polylysine-bilirubin conjugate encapsulated islets remained euglycemic for 35 days PT, while the blood glucose of mice transplanted with untreated islets began to increase at 21 days PT [90]. Few other studies have demonstrated the ability of bilirubin to induce tolerance in islet cell transplantation [121,122]. Considering the challenges of drug delivery for the use of bilirubin in vivo, therapeutic agents that upregulate HO-1 and subsequently bilirubin may be an alternative approach. DMF is a potent antioxidant that may fulfill this role as it has been shown to induce HO-1 in human pancreatic tissue [123]. 

DMF is a potent inducer of the Nrf2 antioxidative system in islet cells. The protective effect of the Nrf2 pathway on islet cells has been extensively studied. We have previously demonstrated that islet cells isolated from rats pretreated with DMF exhibited significantly increased mRNA expression of glutamate-cysteine ligase catalytic subunit (GCLC) and NADPH oxidoreductase 1(NQO1), transcriptional targets of Nrf2, and decreased oxidative stress as indicated by the reduction in 8OHDG positive islet cells [44]. Dh404, another Nrf2 activator, has been shown to improve the viability of islet cells when treated with H_2_O_2_ through the attenuation of oxidative stress by upregulating antioxidative enzymes such as NQO1, HO-1 and GCLC [124]. While there are a plethora of Nrf2 activators that protect islet cells, DMF has been the most evaluated for its immunosuppressive effects. DMF treatment successfully reduce the onset of spontaneous diabetes in NOD mice, a type 1 DM model, suggesting its potent immunomodulatory effects [44]. In fact, DMF has been used in a clinical trial for the treatment of multiple sclerosis (MS), in which DMF treatment significantly decreased both T and B cell counts in MS patients [125]. DMF increased CD4/CD8 and naive/memory T cells while reducing the frequency of Th1 and Th17 inflammatory cells in DMF-treated MS patients [126]. In addition to altering the T cell population, DMF has also been shown to alter M1 (pro-inflammatory)/M2 (anti-inflammatory) macrophage polarization and absolute number. A study that evaluated the effect of DMF on corneal allograft rejection in mice demonstrated that DMF treatment significantly reduced the number of M1 macrophage infiltration into the corneal graft, and DMF inhibited the expression of inflammatory cytokines in macrophages in vitro [127]. Further studies must be conducted to elucidate the potential application of DMF as an immunosuppressant in combination with other agents. In addition to exerting antioxidative and immunosuppressive effects, DMF may also benefit islet cell transplantation through several other aspects. For example, DMF has been found to ameliorate liver ischemia/ reperfusion injury, which, considering that islet cells cause the blockage of small vessels within the liver, is an important approach to improve the safety and recovery after islet cell transplantation [128]. Furthermore, DMF treatment has been shown to suppress proinflammatory cytokine and chemokine production in NOD mice, supporting its anti-inflammatory properties [44]. Lastly, DMF has also been shown to protect against calcineurin inhibitor-induced nephrotoxicity and thus may provide a layer of protection against renal failure after islet cell transplantation [129].

In conclusion, several antioxidants, namely, stem cell-derived EVs, bilirubin, and DMF, have been suggested to protect islet cells from the detrimental effects of oxidative stress while exerting potent immunosuppressive effects through suppression of activation and proliferation of immune cells, alteration in immune cell subtypes, and the induction of regulatory T cells. All in all, a plethora of antioxidants have been evaluated to protect islet cells from oxidative stress throughout the islet cell transplantation process (Table 1). Further studies to elucidate the safety and efficacy of these therapeutic approaches in humans will be vital to assess its application in islet cell transplant.
antioxidants-11-01038-t001_Table 1Table 1Table of antioxidants discussed in this review paper: GSH, reduced glutathione; XOD, xanthine oxidase; GPx, glutathione peroxidase; MnSOD, manganese superoxide dismutase; HO-1, heme oxygenase 1; Nrf2, nuclear erthyroid factor 2. (* DMF has not been evaluated for its immunosuppressive effects in ICT setting, although it has been suggested to exhibit immunomodulation in T1DM).Islet Cell Transplantation StepAntioxidantsMode of ActionSpeciesReferencesProcurement and PreservationUniversity of Wisconsin (UW) solutionROS scavenger (GSH), XOD inhibitionHuman [18,19]Histidine-tryptophan-ketoglutarate (HTK) solutionROS scavenger (Tryptophan)Human [26]Institute Georges Lopez-1 (IGL-1)ROS scavenger (GSH), XOD inhibitionHuman [27]GlutamineROS scavenger (GSH)Rats, Pigs [40,41]MelatoninROS scavengerPigs [43]Ascorbic AcidROS scavengerPigs [43]Dimethyl FumarateNrf2 activatorRats [44]BilirubinInhibition of lipid peroxidationRats [46]Dh404Nrf2 activatorRats [45]Isolation and Processingd-Arg-2’,6’-dimethyltyrosine-Lys-Phe-NH_2_ (SS-31)ROS scavenger, inhibits mitochondrial permeability preventing mitochondrial dysfunctionMice [47]Manganese [III] 5,10,15,20-tetrakis [1,3-diethyl-2imidazoyl] porphyrin (MnTDE)ROS scavengerHuman [48]Glutathione ethyl-ester (GEE)ROS scavenger(GSH)Mice [49]Cell CultureSS-31ROS scavenger, inhibits mitochondrial permeability(prevents mitochondrial dysfunction)Human [47,51]
MnTDEROS scavengerHuman [48]
Curcumin, tetrahydrocurcumin (THC)ROS Scavenger Rat, Mice [61,65]
Glutathione, GEEROS scavenger(GSH)Human [66,67,68]Islet InfusionResveratrol, nobiletinROS scavenger, inhibition of lipid peroxidationHuman, Mice [84,85,86]BilirubinNrf2 activator, activation of GPxMice, Rats [83,87,88,90]Epigallocatechin gallate (ECG)Nrf2 activatorRats [93]Exendin-4 Increase GSH and GPxMice [95]Antiaging glycoproteinUnknownHuman [94]Stem cell derived extracellular vesicles Inreasing GSH and GPxPigs, Human [96,97]Recipient treatment: Immunosuppressivemedication toxicity;ImmunosuppressionAntiaging glycopeptides UnknownHuman [94]GinsengROS scavengerMice, Rats [104,109]CoQ10Protect from mitochondrial dysfunctionRats [105]DDP4 inhibitorsIncreasing MnSODand HO-1/HO-2 expressionRats [106] Exendin-4Increase GSH and GPxRats [95,107]GABA UnknownHuman [108]Bilirubin Nrf2 activator, activation of GPxMice  [90,121,122]Stem cell-derived extracellular vesiclesIncreasing GSH and GPxHuman [110]DMFNrf2 activator
*


## 8. Conclusions

Oxidative stress has been identified as one of the major stressors present throughout the ICT process. Compounding this issue, it has been established that islet cells exhibit significantly reduced expression of antioxidative enzymes, making them exceptionally susceptible to oxidative stress. Thus, there has been an extensive focus on incorporating antioxidants during the preservation, isolation, and culture period in order to improve islet cell viability and function. Furthermore, in more recent years, several of these potent antioxidants have been shown to improve islet cell engraftment, increase vascularization, and even exhibit immunosuppressive properties. Further evaluation of the safety and efficacy of these antioxidants is a crucial step in continuing to improve the overall ICT outcomes. 

## Figures and Tables

**Figure 1 antioxidants-11-01038-f001:**
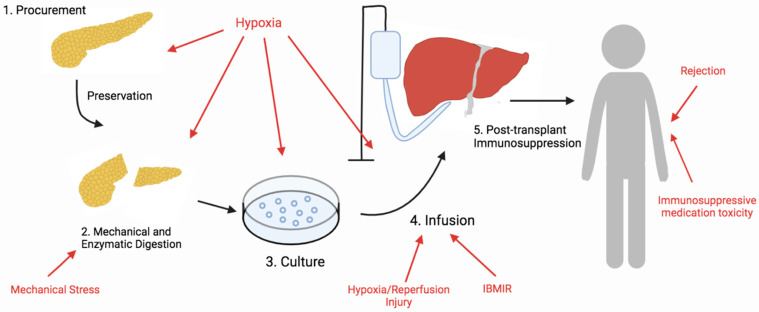
Overview of the steps of islet cell transplantation. Islet cells are exposed to a plethora of stressors that lead to oxidative stress during the islet transplantation process. Sources of oxidative stress during each step are indicated by the red color.

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
