# Peer review of "Anti-Oxidative Therapy in Islet Cell Transplantation"

_antioxidants, 2022, doi:10.3390/antiox11061038_

Round 1

Reviewer 1 Report

The manuscript of Eguchi et al. is aimed at providing an overview of the current status of research and the possible clinical application of anti-oxidant treatments during different stages of the islet transplantation process. The review is structurally well organized and divided into different chapters according to the different stages of islet transplantation. The manuscript is accompanied by 1 figure and one table explaining the sources of ROS during transplantation and summarizing the discussed antioxidants, respectively. Authors cite 116 publications with 13 articles in which the two last authors of the current review participated.

The topic of the review is of interest of the readers and could be a valuable contribution to the journal “Antioxidants”.

There are several major issues however, that need to be addressed before the manuscript could be considered for publication. Indeed, the review provides a very simplistic view of the role of ROS in cell function and fate instead of reflecting the current, more nuanced vision.  A thorough revision of the text is required.

  1. The manuscript focuses on “ROS” in general, without detailing the exact type of ROS produced. This is a common issue with manuscripts concerning oxidants /anti-oxidants. The type of ROS is critical as their elimination involves different anti-oxidant systems [1].

  1. The manuscript only deals with oxidative stress, failing to address the opposite scenario, the onset of reductive stress. The notion of redox imbalance (not just oxidative stress) should be better explored by the review (see for example [2; 3]).

  1. ROS are important mediators of cellular signaling. Indiscriminative elimination of ROS hampers these critical signaling pathways and to some extent, can explain the failure of anti-oxidant therapies in clinical settings [4].

  1. Page 2 Line 43: “Considering that human β cells exhibit profound deficiency in antioxidant capacity,….”

This statement is only partially correct. Beta cells are poor in Mn-SOD, glutathione peroxidase, and catalase but have higher levels of glutaredoxin and peroxiredoxins. Authors should correct this sentence with the appropriate citation.

  1. Table 1. It should be noted whether the study has been conducted in rodents/pigs/ human islets.

  1. An additional Table summarizing the mode of actions of the noted antioxidants should be presented.

Minor issue

  1. Some of the abbreviations are not explained e.g. HO-1, NQO1. Authors should check the manuscript for additional examples.
  2. Custodiol-N composition is not explained.

    References

    [1]       Forman, H.J., Zhang, H., 2021. Targeting oxidative stress in disease: promise and limitations of antioxidant therapy. Nat Rev Drug Discov 20(9):689-709.

    [2]       Yan, L.J., 2014. Pathogenesis of chronic hyperglycemia: from reductive stress to oxidative stress. J Diabetes Res 2014:137919.

    [3]       Xiao, W., Loscalzo, J., 2020. Metabolic Responses to Reductive Stress. Antioxid Redox Signal 32(18):1330-1347.

    [4]       Sies, H., Belousov, V.V., Chandel, N.S., Davies, M.J., Jones, D.P., Mann, G.E., et al., 2022. Defining roles of specific reactive oxygen species (ROS) in cell biology and physiology. Nat Rev Mol Cell Biol.

Author Response

We are grateful to the editor and the reviewers for the positive overall feed-back and for the helpful criticism.  Our revised paper is much improved, and we are hopeful that it now fulfils the quality standards needed for acceptance on Antioxidants. 

Reviewer 1

The manuscript of Eguchi et al. is aimed at providing an overview of the current status of research and the possible clinical application of anti-oxidant treatments during different stages of the islet transplantation process. The review is structurally well organized and divided into different chapters according to the different stages of islet transplantation. The manuscript is accompanied by 1 figure and one table explaining the sources of ROS during transplantation and summarizing the discussed antioxidants, respectively. Authors cite 116 publications with 13 articles in which the two last authors of the current review participated.

The topic of the review is of interest of the readers and could be a valuable contribution to the journal “Antioxidants”. 

There are several major issues however, that need to be addressed before the manuscript could be considered for publication. Indeed, the review provides a very simplistic view of the role of ROS in cell function and fate instead of reflecting the current, more nuanced vision.  A thorough revision of the text is required.

  1. The manuscript focuses on “ROS” in general, without detailing the exact type of ROS produced. This is a common issue with manuscripts concerning oxidants /anti-oxidants. The type of ROS is critical as their elimination involves different anti-oxidant systems [1].

Thank you for your valuable suggestion. In the section “ Islet cells and oxidative stress” we discussed the primary sources of ROS production in islet cells. Previous articles discussed in this review did not specify which ROS was generated under different conditions or at each stage of islet isolation process.  They typically evaluated “Reactive Oxygen Species” as a whole or used indirect measures of oxidative stress such as 8OHDG. All specific ROS information we could find was added into the appropriate sections.

  1. The manuscript only deals with oxidative stress, failing to address the opposite scenario, the onset of reductive stress. The notion of redox imbalance (not just oxidative stress) should be better explored by the review (see for example [2; 3]).

Thank you for your insightful comment. We added a paragraph in “Islet cells and oxidative stress” discussing reductive stress as a potential limiting factor of antioxidative therapy. In our knowledge, very little research has been done regarding the effects of redox imbalance on islet transplantation.

  1. ROS are important mediators of cellular signaling. Indiscriminative elimination of ROS hampers these critical signaling pathways and to some extent, can explain the failure of anti-oxidant therapies in clinical settings [4].

Thank you for your valuable suggestion. We strongly agree your comments. In fact, we were aware that higher level of anti-oxidative therapies is harmful for β cell function and/or survival in our experimental animal models. We believe that proper treatment dose and timing are critical for successful anti-oxidant therapies in clinical settings. We added a paragraph in “Islet cells and oxidative stress” discussing the importance of ROS in insulin signaling.

  1. Page 2 Line 43: “Considering that human β cells exhibit profound deficiency in antioxidant capacity,….” This statement is only partially correct. Beta cells are poor in Mn-SOD, glutathione peroxidase, and catalase but have higher levels of glutaredoxin and peroxiredoxins. Authors should correct this sentence with the appropriate citation.

Thank you for your comment. We talked more in detail about the antioxidative system in islet cells in the section “ Islet cells and oxidative stress”

  1. Table 1. It should be noted whether the study has been conducted in rodents/pigs/ human islets.

Thank you for your comment. We added a column specifying species.

  1. An additional Table summarizing the mode of actions of the noted antioxidants should be presented.

Thank you for your comment. We added another column specifying mechanism of action according to the papers presented in this paper.

Minor issue

  1. Some of the abbreviations are not explained e.g. HO-1, NQO1. Authors should check the manuscript for additional examples.

Thank you for your comment. We have gone through and made sure all abbreviations are explained.

  1. Custodiol-N composition is not explained.

Thank you for your comment. Which added a reference with the composition of Custodiol N in the review.

References

[1]       Forman, H.J., Zhang, H., 2021. Targeting oxidative stress in disease: promise and limitations of antioxidant therapy. Nat Rev Drug Discov 20(9):689-709.

[2]       Yan, L.J., 2014. Pathogenesis of chronic hyperglycemia: from reductive stress to oxidative stress. J Diabetes Res 2014:137919.

[3]       Xiao, W., Loscalzo, J., 2020. Metabolic Responses to Reductive Stress. Antioxid Redox Signal 32(18):1330-1347.

[4]       Sies, H., Belousov, V.V., Chandel, N.S., Davies, M.J., Jones, D.P., Mann, G.E., et al., 2022. Defining roles of specific reactive oxygen species (ROS) in cell biology and physiology. Nat Rev Mol Cell Biol.

Reviewer 2 Report

This review discussed the potential application of antioxidative therapy during each step of islet cell transplantation. This review provides a good alternative solution to the problems existing in islet cell transplantation. The summary is relatively standard in writing, but there are two suggestions. I hope the author can improve it.

  1. The author introduces different antioxidant methods from different stages of islet cell transplantation, and these antioxidants are almost different at different stages.Oddly enough, do antioxidants exhibit such time specificity? It is suggested that the author also summarizes the effects of the same antioxidant on different stages.
  2. Although the antioxidant capacity of human islet cells is relatively poor, the current antioxidant substances are only one of the solutions to the problem of islet transplantation. Is there any other solution? In other words, in addition to antioxidation, what mechanism does the antioxidant substances play? I hope the author can supplement it.

Author Response

We are grateful to the editor and the reviewers for the positive overall feed-back and for the helpful criticism.  Our revised paper is much improved, and we are hopeful that it now fulfils the quality standards needed for acceptance on Antioxidants. 

Reviewer 2

This review discussed the potential application of antioxidative therapy during each step of islet cell transplantation. This review provides a good alternative solution to the problems existing in islet cell transplantation. The summary is relatively standard in writing, but there are two suggestions. I hope the author can improve it.

1. The author introduces different antioxidant methods from different stages of islet cell transplantation, and these antioxidants are almost different at different stages. Oddly enough, do antioxidants exhibit such time specificity? It is suggested that the author also summarizes the effects of the same antioxidant on different stages.

Thank you very much for your insightful comments. We also believe that the discussed antioxidants would not be stage specific and would work for most stages of islet cell transplantation. Therefore, we went back to check whether there were overlap in any other antioxidants and found two more Bilirubin for procurement/ isolation and exendin-4 for islet infusion.

2. Although the antioxidant capacity of human islet cells is relatively poor, the current antioxidant substances are only one of the solutions to the problem of islet transplantation. Is there any other solution? In other words, in addition to antioxidation, what mechanism does the antioxidant substances play? I hope the author can supplement it.

Thank you very much for your valuable comment. We included anti-inflammatory properties for Bilirubin and DMF in addition to the discussed benefits of vascularization and immunosuppressive effects in sections “antioxidative treatment during islet infusion” and “antioxidative treatments in islet graft recipients.

Round 2

Reviewer 1 Report

Authors provided the recommended changes to the manuscript and replied to the reviewer's comments in a satisfactory manner.